

# Hadronic footprint of GeV-mass dark matter

Tilman Plehn[1], Peter Reimitz[1*] and Peter Richardson[2,3]

**1** Institut für Theoretische Physik, Universität Heidelberg, Germany
**2** Theoretical Physics Department, CERN, Geneva, Switzerland
**3** Institute for Particle Physics Phenomenology, Durham University, UK

★ reimitz@thphys.uni-heidelberg.de

## Abstract

GeV-scale dark matter is an increasingly attractive target for direct detection, indirect detection, and collider searches. Its annihilation into hadronic final states produces a challenging zoo of light hadronic resonances. We update Herwig7 to study the photon and positron spectra from annihilation through a vector mediator. It covers dark matter masses between 250 MeV and 5 GeV and includes an error estimate.

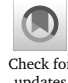
## Content

## 1 Introduction

The fundamental nature of dark matter is the biggest particle physics question of our time. It follows directly from the success of quantum field theory in describing the properties of elementary particles, as well as the standard cosmology after the Big Bang. A problem is that the term particle dark matter is very loosely defined, covering very light particles essentially

giving a background wave function all the way to primordial black holes. Theory motivations are widely used to support certain mass ranges, but given the generally modest success of models for physics beyond the Standard Model they should be taken with a truck load of salt [1].

The defining feature of dark matter particles — closely related to the one actual measurement of the relic density [2] — is the dark matter production mechanism. It needs to explain the observed relic density in agreement with the largely known thermal history of the universe. For wide classes of dark matter models this implies an interaction with SM particles beyond an obviously existing gravitational interaction. Dark matter masses around the weak scale and down to the GeV scale can be produced thermally through freeze-out or freeze-in, or through a relation with the baryon asymmetry in the universe. These mechanisms require more or less strong couplings to the SM matter particles, *i.e.* to leptons or quarks. In this mass range there exists a wealth of relatively model-independent direct detection constraints [3], and their extension to lighter dark matter particles at and below the GeV-scale is one of the most interesting experimental directions [4–6]. For such GeV-scale dark matter especially the couplings to quarks are not well constrained.

Indirect searches for dark matter are another way to directly probe the properties of the dark matter in the universe. A leading signature are photons produced in the annihilation of dark matter in dense regions of the sky [7–10]. These photons can constrain dark matter interactions with the Standard Model on the lepton and the hadron side. The reason is that they can be produced directly and through radiation from any charged annihilation product. Just as for positrons, we know the photon spectrum from many particle physics experiments over the recent decades. They are available for instance through the PPPC4DMID tool [11] based on PYTHIA [12]. Standard tools like MICROMEGAS [13, 14], MADDM [15, 16], or DARK-SUSY [17, 18] include similar spectra based on multi-purpose Monte Carlo generators. A major technical problem with dark matter annihilation into hadrons is that its description is not available through PYTHIA once the dark matter masses drop below around 5 GeV. The only exception is the recent HAZMA [19] tool for dark matter masses below 250 MeV [20]. This leaves dark matter annihilation to the leading hadronic final states for masses between 250 MeV and 5 GeV essentially uncovered.

Technically, GeV-scale dark matter annihilation through light scalar or light vector mediators is very different. If we assume that a new scalar couples to SM particles with Yukawa couplings roughly reflecting the SM mass hierarchy, increasingly weak GeV-scale dark matter will annihilate into charm quarks and tau leptons, followed by muons and eventually pions and electrons. From a hadronic physics point of view the more interesting scenario are vector mediators, where the SM interactions are generation-universal. In that case we will observe a wealth of hadronic annihilation channels below the $b\bar{b}$ threshold. These annihilation channels will have distinct photon and lepton spectra, which we will focus on in this study.

Finally, the proper description of dark matter annihilation to hadronic final state is plagued by large uncertainties, as for instance pointed out for PPPC4DMID [11] in relation to PYTHIA [12] and HERWIG [21]. Consequently, dedicated comparisons between PYTHIA and HERWIG have been published for dark matter annihilation to tau leptons, bottom and top quarks, and weak bosons [22]. More recently, this comparison has been updated [23] to the most recent versions of PYTHIA8 [12, 24] and HERWIG7 [25, 26]. A detailed analysis of the the PYTHIA predictions can be found in Ref. [27]. All of these studies target relatively heavy dark matter annihilation, in line with the common weakly interacting massive particle (WIMP) hypothesis.

In this paper we provide the first proper description of photon and lepton spectra from GeV dark matter annihilating into hadronic final states based on HERWIG with an updated

fit to electron–positron data, including several new final states. They become relevant when we reduce the dark matter scattering energy below the PYTHIA limit. We update the fit to electron–positron data as the input to the HERWIG description and add the necessary new hadronic final states with up to four hadrons. Especially for the photon spectrum we observe a complete change in the shape of the spectrum when we reduce the dark matter mass, starting from typical hadron decay chains to continuum multi-pion production. In addition we provide a first estimate of the impact of the input-data fit uncertainties on the output spectra.

The paper is structured the following way: after introducing our toy model in Sec. 2 we review the established implementations in Sec. 3. We show how their reliability starts to fade once we go below dark matter masses of 5 GeV and the tools start to extrapolate beyond their common PYTHIA input. In Sec. 4 we show the results from our new HERWIG-based implementation. We focus on shape changes in the photon and lepton spectra when we reduce the dark matter mass towards into the continuum-pion regime. We also show the error bands on the photon and positron spectra from the fit uncertainties to the electron-positron input data. In the Appendix we provide all details about our new fit, the underlying parametrizations, the best-fit points, and the error bands.

## 2 Toy model

The standard interpretation framework for weak-scale dark matter is thermal freeze-out production or the WIMP paradigm. Embedding GeV-scale dark matter searches in a global analysis [28] provides an excellent illustration of the many cosmological constraints and their model dependence. Above masses around 10 GeV, FERMI constrains these scenarios using photons in dwarf spheroidal galaxies [29,30], while AMS covers leptonic final states [31,32]. The E-ASTROGAM program [33], for example, is proposing searches for gamma-rays in the MeV-GeV region. In addition, precision measurements of the Cosmic Microwave Background (CMB) [2] are sensitive to the total ionizing energy either directly (electrons and muons) or indirectly. Finally, Big-Bang Nucleosynthesis (BBN) does not allow WIMPs below around 10 MeV [34,35]. The main difference between these different analyses is that some rely on assumptions on the thermal history of dark matter.

For DM masses in the range $m_\chi = 0.1 \dots 7$ GeV the CMB provides the leading constraint on GeV-scale dark matter, where asymmetric dark matter as an alternative production model leads to weaker CMB constraints if the dark matter is sufficiently asymmetric [36]. For an anti-DM to DM ratio of less than $\sim 2 \times 10^{-6}(10^{-1})$ for DM masses $m_\chi = 1$ MeV($10$ GeV), CMB constraints can be evaded as seen in Fig. 1 of [36]. Not yet being fully asymmetric, one still gets indirect detection signals. Other modifications at least weakening the CMB constraints for thermal production are softer spectra from annihilation modes beyond $2 \rightarrow 2$ kinematics [37, 38], including a dominant $2 \rightarrow 3$ bremsstrahlung process [39–54]. However, the necessary annihilation rate is typically too small to lead to observed relic density.

To define a toy model for our hadronization study we assume that the observed dark matter density is somehow produced through thermal freeze-out, but with a light vector mediator. We assume the dark matter candidate to be a Majorana fermion $\chi$, but our results apply the same way to asymmetric dark matter where the dark matter fermion has to be different from its anti-particle. Since our study is based on $e^+e^-$- data, we have to focus on vector mediator models. A simple mediator choice starts from an additional $U(1)$ gauge symmetry, where we gauge one of the accidental global symmetries related to baryon and lepton number [55–59]. For our purpose of testing dark matter annihilation into light-flavor jets with a limited number

of photons from leptonic channels the most attractive combination is $B - 3L_\mu$ [60]. This gives us the annihilation channel

$$\chi\chi \to Z' \to \text{Standard Model}. \tag{1}$$

To avoid strong biases from an underlying model we also show results for $Z'$ couplings similar to the Standard Model case for low energies. For consistent field theory models the annihilation to SM quarks will always occur at the loop level, even if they are suppressed at tree level [61]. As our benchmark model we therefore assume an approximately on-shell annihilation

$$\chi\chi \to Z' \to q\bar{q} \qquad \text{with} \quad m_{Z'} \approx 2m_\chi. \tag{2}$$

The coupling strength of the DM to the mediator can be chosen arbitrarily, because we are only interested in the form of the energy spectra from the hadronic final states. For light dark matter masses the relevant quarks are $u, d, s, c$, possibly the bottom quark. The charm quark plays a special role, because threshold region is poorly understood. Examples for distinct photon spectra from annihilations to $c$ and $b$ quarks are, for example, described in [62]. All we can do is rely on the spectra included in PYTHIA or HERWIG, with little improvement on the modelling side.

For the three lightest quarks there exists a wealth of measurements which we can use to constrain dark matter annihilation into hadrons. We decompose a quark DM current $J_{\text{DM}}^\mu = \sum_{q=u,d,s} a_q \bar{q}\gamma^\mu q$ into isospin components and a separate $s\bar{s}$ contribution,

$$J_{\text{DM}}^\mu = \frac{1}{\sqrt{2}} \left( (a_u - a_d) J^{I=1,3,\mu} + (a_u + a_d) J^{I=0,\mu} \right) + a_s J^{s,\mu}, \tag{3}$$

where $a_q$ are the couplings of the light vector mediator to the light quarks, $q = u, d, s$. The mediator couplings to quarks are fixed to $a_q = 1/3$ for any anomaly-free $B - L$ model. Depending on the mediator coupling structure to quarks, one or the other isospin current might vanish. As a consequence, some resonance contributions to the channels might vanish, or even more drastically, pure isospin $I = 0$ channels, for example

$$\chi\chi \to \omega\pi\pi, \eta\omega, \dots \tag{4}$$

or pure $I = 1$ channels such as

$$\chi\chi \to \pi\pi, 4\pi, \eta\pi\pi, \omega\pi, \phi\pi, \eta'\pi\pi, \dots \tag{5}$$

are absent. We also choose to include the isospin breaking contribution from $\omega \to \pi^+\pi^-$ in the $I = 1$ current for simplicity. The general matrix element for DM annihilation can be written in the form

$$\mathcal{M} = a_{\text{DM}} \bar{v}(p_1)\gamma^\nu u(p_2) d_{\nu\mu}^{\text{DM}} \langle X | J_{\text{DM}}^\mu | 0 \rangle, \tag{6}$$

with the DM-mediator coupling $a_{\text{DM}}$ and the vector mediator propagator $d_{\nu\mu}^{\text{DM}}$.

In our toy model we always assume $m_{Z'} = 2m_\chi$, but given the non-relativistic nature of DM annihilation the mediator mass should only have negligible impact on our spectra. Since the mass of the mediator determines the width of the mediator, we calculate the width in the hadronic resonance region within HERWIG through its decays to all kinematically allowed hadronic final states listed in Tab. 2 of the Appendix.

# 3 Established tools

Different public tools generate energy spectra for different DM annihilation channels to SM particles. They are limited in DM masses by their approach and by their back-end, but are mediator-independent. We summarize

- PPPC4DMID [11] provides tabulated energy spectra for indirect detection. The $e^{\pm}$, $\bar{p}$, $\bar{d}$, $\gamma$, and $\nu_{e,\mu,\tau}$ fluxes are generated with PYTHIA8.135 [12] down to $m_\chi = 5$ GeV. We use the provided interpolation routine to extrapolate the results to $m_\chi = 2$ GeV.

- MICROMEGAS [13,14] uses tabulated PYTHIA spectra for $\gamma, e^+, \bar{p}, \nu_{e,\mu,\tau}$ and extrapolates down to $m_\chi = 2$ GeV. In the manual of version MICROMEGAS2.0 it is mentioned that the strategy for calculating spectra is analogous to that of DARKSUSY and that spectra extrapolated to masses below 2 GeV should be taken with care.

- MADDM [15, 16] provides two ways of calculating the energy spectra both based on PYTHIA [24]. The 'fast' calculation is based on the numerical tables provided by PPPC4DMID. In the 'precise' mode, events are generated with MADGRAPH and then passed to PYTHIA for showering and hadronization. In this mode it is possible to calculate the fluxes of any final states based on the UFO model implementation.

- DARKSUSY [17,18] provides tables down to 3 GeV for energy spectra of two-particle SM final states based on PYTHIA6.426 [63]. The tool can interpolate and extrapolate the $\gamma, e^+, \bar{p}, \bar{d}, \pi_0, \nu_{e,\mu,\tau}, \mu$ fluxes for all quark final states. In addition it includes annihilation to $\mu\mu$, $\tau\tau$, gluons, and weak bosons. Dark matter annihilation into $e^+e^-$ pairs appears to not be included.

- HAZMA [19] is a Python toolkit to produce energy spectra in the sub-GeV range. It is based on leading order chiral perturbation theory and is valid in the non-resonance region below $m_\chi = 250$ MeV.

From this list it is clear that for dark matter masses in the GeV range all public tools are based on PYTHIA, one way or another. Multi-purpose Monte Carlo tools, such as, PYTHIA or HERWIG can calculate the energy spectra for many hard scattering processes, followed by hadronization or fragmentation and hadron decays. In the range we are interested in these spectra are usually extracted from data, as discussed in the Appendix.

The advantage of the Monte Carlo tools is that we can extract the cosmologically most relevant photon, lepton, and anti-proton spectra for each hard dark matter annihilation process. We assume an annihilation process of the kind given in Eq.(2), but allow for any kinematically allowed SM final state. For the numerical results we rely on spectra from the processes

Table 1: Comparison of publicly available tools to generate spectra from DM annihilation.

| Tool | Back-end | $m_\chi^{\min}$ | DM models |
|---|---|---|---|
| PPPC4DMID | PYTHIA8.135 tables | 5 GeV | generic DM |
| MICROMEGAS | PYTHIA6.4 tables | $\sim 2$ GeV | UFO model |
| MADDM | PPPC4DMID PYTHIA8.2 direct | 5 GeV $\sim 2$ GeV | UFO model |
| DARKSUSY | PYTHIA6.426 tables | $\sim 3$ GeV | generic DM, SUSY |

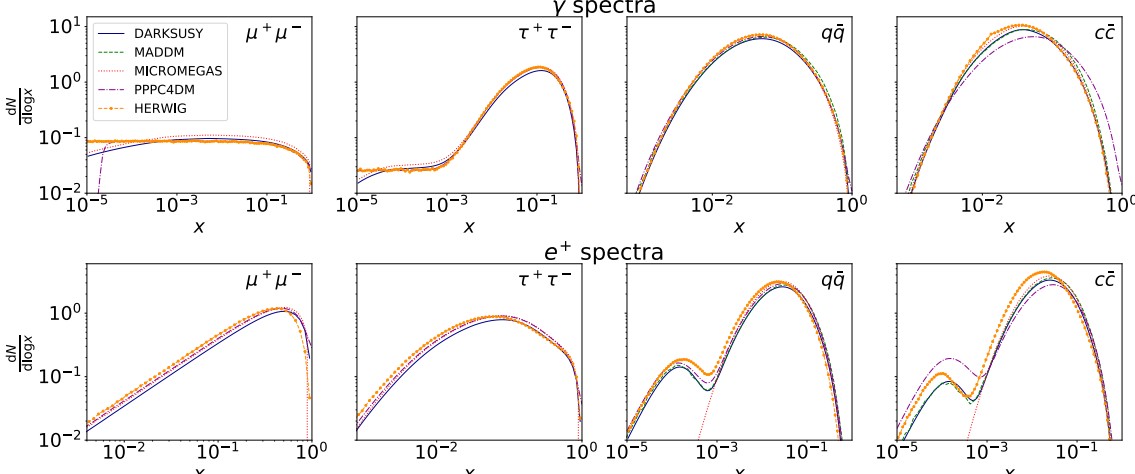

Figure 1: Photon and positron spectra $dN/d\log x$ with $x = E_{\rm kin}/m_\chi$ for $m_\chi = 5$ GeV from the different hard annihilation channels. We show results from DARKSUSY, MADDM, MICROMEGAS, PPPC4DMID, and HERWIG.

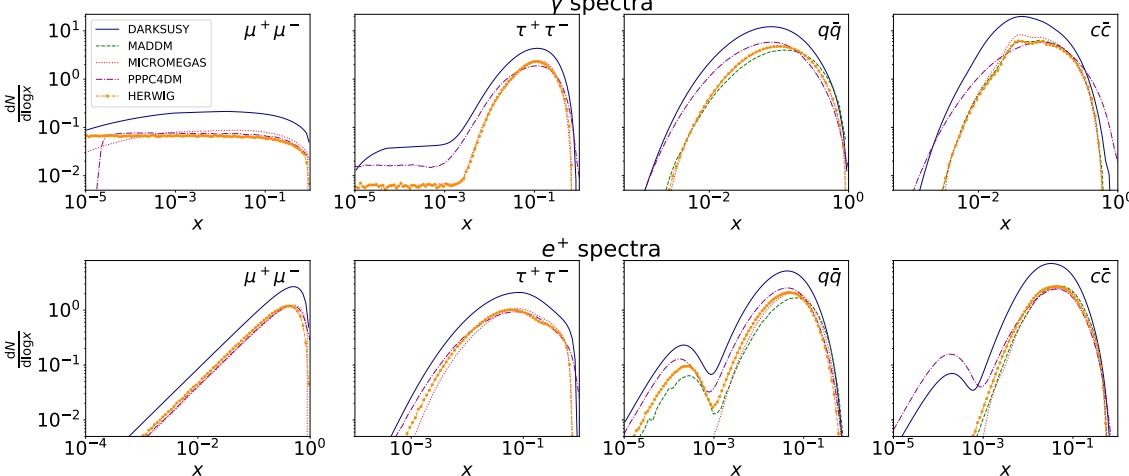

Figure 2: Photon and positron spectra $dN/d\log x$ with $x = E_{\rm kin}/m_\chi$ for $m_\chi = 2$ GeV from the different hard annihilation channels. We show results from DARKSUSY, MADDM, MICROMEGAS, PPPC4DMID, and HERWIG.

$e^+ e^- \to$ SM pairs, at a given energy $m_{ee} = 2m_\chi$. In Fig. 1 we compare the corresponding PYTHIA-like spectra from the standard tools discussed above. We show the photon and positron spectra from DM annihilation into muon, tau, and light-quark $(u, d, s)$ pairs and compare them to the standard HERWIG output for an alternative description. Starting with the left panels of Fig. 1 we see a flat photon spectrum from soft-enhanced radiation and a triangular positron spectrum from the $\mu^+$-decay with a three-particle final state. For taus the hadronic decays produce neutral and charged pions, where for instance the decay $\pi^0 \to \gamma\gamma$ dominates down to $x \approx 10^{-3}$. Below that we again find the flat photon spectrum from soft emission. The dominant contribution to the position spectrum is the hadronic decay chain $\tau^+ \to \pi^+ \to \mu^+ \to e^+$, with a sub-dominant contribution from the leptonic $\beta$-decay $\tau^+ \to e^+$. Next, light-flavor quarks $u, d, s$ form a range of hadrons which then decay to $\pi^0 \to 2\gamma$. The positron spectrum from these light quarks includes a soft neutron $\beta$-decay, which gives rise to the secondary maximum around $x \approx 10^{-4}$. The neutron decay is not included in our default version of MICROMEGAS, but can be easily added. Finally, moving to DM annihilation into

charms we see that the photon and positron spectra are the same as for the light quarks.

In Fig. 2 we show the same spectra, but for a slightly lower dark matter mass of 2 GeV. This value is slightly beyond where PYTHIA output can be used in a straightforward manner. Essentially all radiation and decay patterns remain the same as for 5 GeV, but the different curves start moving apart. This is an effect of individual extrapolations from the PYTHIA output. The only interesting feature appears in the annihilation $\chi\chi \to c\bar{c}$. Here the extrapolated results from PPPC4DMID and DARKSUSY still include a secondary peak corresponding to the neutron decay in the light quark channel. However, the lightest charm baryon is $\Lambda_c$ has a mass of 2.29 GeV, so at $m_\chi = 2$ GeV it cannot be produced on-shell. What we see is likely an over-estimate of off-shell effects or an extrapolation error from the 5 GeV case, which illustrates the danger of ignoring the explicit warning not to use for instance PPPC4DMID or DARKSUSY below their recommended mass ranges. For MICROMEGAS the spectrum is significantly softer than from the dedicated MADDM call to PYTHIA and from HERWIG.

Altogether we find that for $m_\chi = 5$ GeV there is a completely consistent picture, where the PYTHIA-based results are in excellent agreement with HERWIG. Going to $m_\chi = 2$ GeV leads to an increased variation between the different tools and illustrates why we might not want to use the standard tools outside their recommended mass ranges.

## 4 Herwig4DM spectra

To extend the range of valid simulations of dark matter annihilation to quarks we start with the standard HERWIG7 [25, 26, 64] implementation. We then add a set of additional final states and update some other spectra, as discussed in the Appendix. This allows us to cover DM masses down to twice the pion mass for vector mediator models. Below the threshold $m_{Z'} = 2m_\pi \approx 250$ MeV the annihilation to hadrons will be suppressed and the annihilation to electrons and photons will dominate. In Fig. 3 we show the photon and positron spectra from the annihilation process

$$\chi\chi \to Z' \to q\bar{q}, \qquad \text{with} \qquad q = u, d, s, c \qquad (7)$$

for decreasing DM masses from $m_\chi = 2$ GeV to 250 MeV.

**Spectra**

Most photons and positrons in hadronic processes come from neutral and charged pion decays, respectively. These pions are either directly produced or are the end of a decay chain of all forms of hadronic states listed in Tab. 2 in the Appendix. In a few cases, photons can also be directly produced in DM annihilation, for instance

$$\chi\chi \to \eta\gamma, \pi\gamma \,. \qquad (8)$$

In the left panel of Fig. 3 we see how photon production channels drop out when we reduce the DM mass or center-of-mass energy of the non-relativistic scattering process. Whereas for $m_\chi > 1$ GeV all possible hadronic final states contribute to the round shape of the spectrum, for lower energies only photons and positrons from very specific processes give a characteristic energy spectrum.

For example for $m_\chi = 500$ MeV or equivalently a center-of-mass energy of 1 GeV we expect two kaons from the $\phi$ resonance to provide most photons through consecutive decays of kaons to pions to photons. This leads to a triangular shape of the photon spectrum. If we go down to 250 MeV, the only remaining annihilation channels are

$$\chi\chi \to \pi^0\gamma, \pi\pi, 3\pi \,. \qquad (9)$$

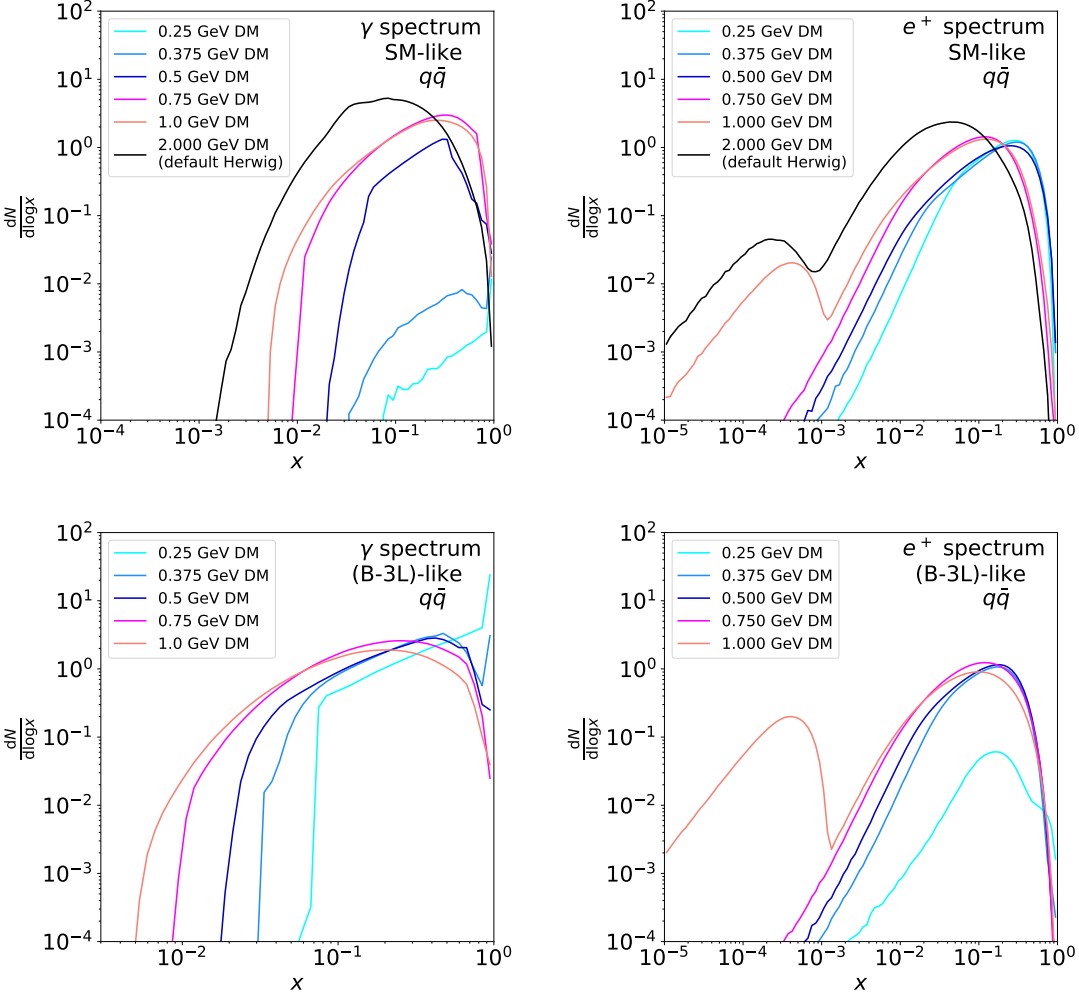

Figure 3: Photon and positron spectra $dN/d\log x$ with $x = E_{\text{kin}}/m_\chi$ for $m_\chi = 0.25\ldots 2$ GeV from $u, d, s, c$ quarks with SM-like and $(B-3L)$-like couplings. We use our modified version of HERWIG7 for all curves below 2 GeV.

Of those, the photons mainly come from the $\pi^0\gamma$ final state, so one photon is produced directly with an energy around the DM mass. It leads to the sharp peak around $x \approx 1$. The additional photons come from the $\pi^0$-decay and are responsible for the distribution to roughly $x \approx 10^{-1}$. The same applies for $m_\chi = 375$ MeV with an additional bump-like contribution from neutral pions in the $3\pi$ and $4\pi$ channels as well as additional photons from the dominantly neutral $\eta\gamma \to (2\gamma)\gamma, (3\pi^0)\gamma$ decay including a direct photon.

The basic shape of the positron spectrum is given by the neutron pair production threshold. Above threshold, we observe an additional peak slightly above $x \sim 10^{-4}$ resulting from positron production in the neutron $\beta$-decay. For $m_\chi < 1$ GeV, all positrons come from charged pion decays. The peak position depends on how early that charged pion decay occurs for the dominant processes at the respective center-of-mass energy. For example, for $m_\chi < 500$ MeV, charged pions are mainly produced directly in $\pi\pi, 3\pi, 4\pi$ production and hence the peak of the spectrum is shifted towards $x = 1$.

As mentioned in Sec. 2, the composition of the DM current changes with the way the mediator couples to quarks. In any $(B-3L)$-like model with equal couplings to quarks, the

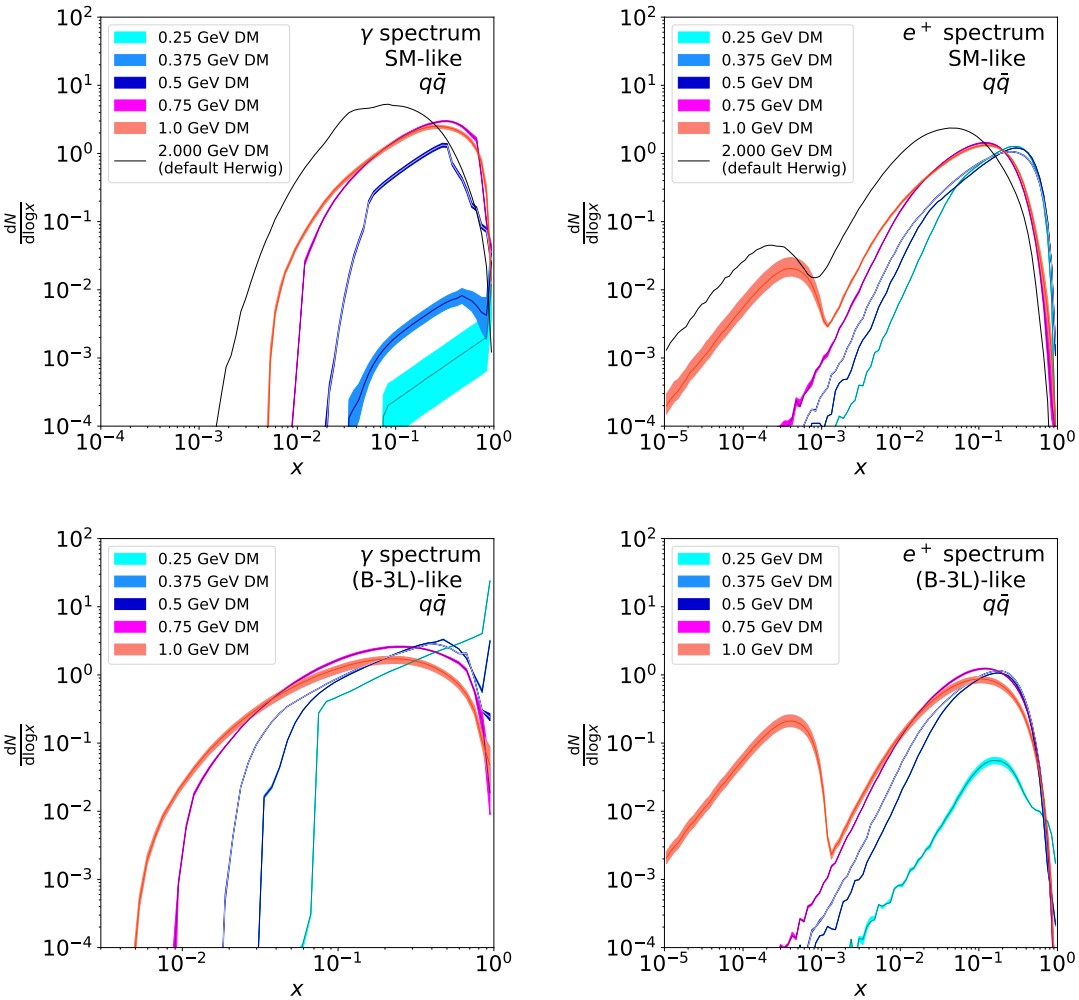

Figure 4: Photon and positron spectra $dN/d\log x$ with $x = E_{\text{kin}}/m_\chi$ for $m_\chi = 0.25 \dots 2$ GeV from $u, d, s, c$ quarks with SM-like and $(B - 3L)$-like couplings with uncertainty bands allowing for perfect cancellations. The 2 GeV curve and the central values correspond to Fig. 3.

isospin $I = 1$ contribution vanishes and consequently some resonance contributions as well as all channels listed in Eq.(5) vanish. For $m_\chi = 250$ MeV this implies that without the $\pi\pi$ channel, $\pi^0\gamma$ becomes the dominant annihilation mode. The direct photon production lifts the photon spectrum, as seen in the upper panels of Fig. 3. This is accompanied by a drop in the positron spectrum that only receives contributions from the subdominant $3\pi$ final state. If we choose a center-of-mass energy below the $3\pi$ threshold, positron spectra from quarks would be completely absent. For $m_\chi = 375$ MeV with an increasing $3\pi$ contribution towards the $\omega(782)$ resonance, the position spectra are lifted. For higher energies and the contribution from several channels, the $(B - 3L)$-like spectra resemble the SM-like case. Although their sources are not identical channel by channel, the way the photons and positions are produced is similar.

## Error bands

Uncertainties on the energy spectra are dominated by the uncertainties from the fits to electron-positron data discussed in the Appendix. We define ranges of model parameters to cover bands

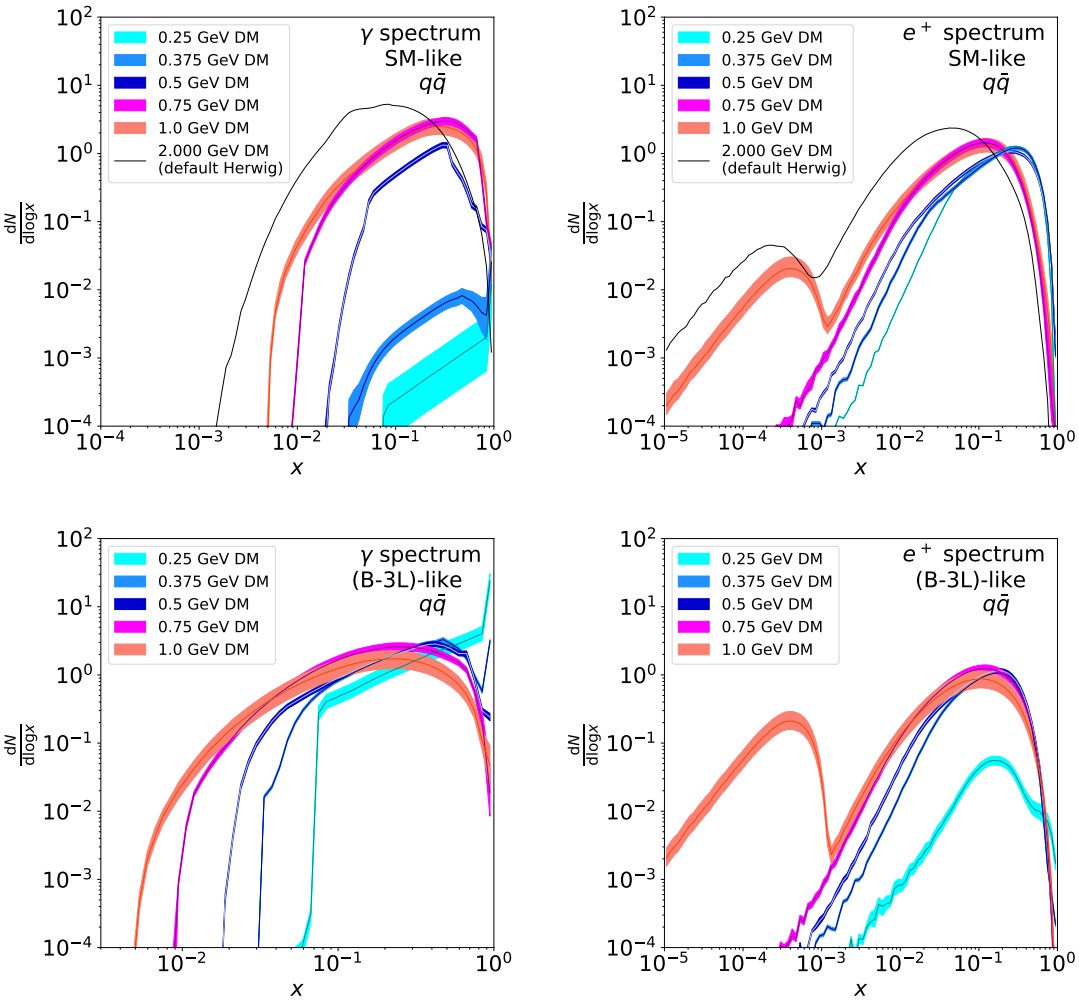

Figure 5: Photon and positron spectra $dN/d\log x$ with $x = E_{\text{kin}}/m_\chi$ for $m_\chi = 0.25 \dots 2$ GeV from $u, d, s, c$ quarks with SM-like and $(B-3L)$-like couplings with very conservative uncertainty bands. The 2 GeV curve and the central values correspond to Fig. 3.

in the $e^+e^-$-annihilation cross sections as a function of the energy and propagate those parameter ranges through the hadronic currents into the energy spectra. This means that the error on a given spectrum corresponds to the uncertainty of the dominant channel at the corresponding energy.

In the upper panels of Fig. 4 we see that the photon spectrum at $m_\chi = 250$ MeV inherits large uncertainties from the poorly measured dominant $\pi^0\gamma$ channel in that energy range. For $m_\chi = 375$ MeV the more precisely measured $3\pi$ channel suppresses the $\pi^0\gamma$ channel, but still leaves us with visible error bands. For even higher energies several channels contribute to the uncertainty of the photon spectrum. We observe the smallest error bands for spectra that benefit from precisely measured dominant processes, for instance peak regions such as the $\phi$ resonance at 1 GeV in the $KK$ channel, the $\rho$ resonance in the $2\pi$ decay, or generally well-measured channels such as $4\pi$. Positron spectra with their dominant $2\pi, 3\pi, 4\pi$ channels are always well measured. The only exception is $m_\chi = 1$ GeV spectrum, especially the lower peak around $\sim 10^{-4}$, which comes from the neutron $\beta$-decay. As discussed in the Appendix, the $n\bar{n}$ channel is poorly measured and leaves us with larger uncertainties in that regime.

In $(B-3L)$-like models, we will not get any contributions from well-measured $2\pi$ and $4\pi$ final states. This means the uncertainties on the position spectrum for $m_\chi = 250$ MeV are slightly larger than in the SM-like case, see the lower panel of Fig 4. Nevertheless, as long as no channel drops out and another channel with larger uncertainties starts to dominate, the uncertainties in the $(B-3L)$-like case tend to be smaller. The reason is the absence of the $I = 1$ contributions and their sizeable uncertainties.

Finally, we want to ensure that our error estimates are conservative. In Fig. 4 we use the uncertainties on the individual channels bin-wise, add all contributions up and normalize by the sum of their corresponding cross-sections. For channels with large cross-sections that are also giving the main contribution to the total amount of photons/positrons in the spectrum, the error bars can completely cancel for the normalized spectra. This way, we only get sizable uncertainty bands for spectra where one channel is dominating the shape of the spectrum, but is playing a sub-dominant role in the total cross-section. An example is the $\pi^0\gamma$ final state for the SM-like photon spectrum at $m_\chi = 250$ MeV or the lower bump in the 1 GeV positron spectrum caused by $n\bar{n}$. This assumption can be considered somewhat aggressive in a situation where we do not have full control of the full error budget. Instead, we can maximize and minimize all spectra channel by channel and separately normalize them by the smallest and largest total cross-section possible. This way there will be no cancellation for single-channel spectra, and in Fig. 5 we indeed see much increased uncertainties. Obviously, the real error bands are going to be somewhere between the results shown in Fig. 4 and Fig. 5 determined by analysis details beyond the scope of this first analysis.

## 5 Outlook

We have studied the positron and photon spectra from non-relativistic dark matter annihilation in a dark matter mass range from 250 MeV to 5 GeV (with the exception of the poorly understood region near the charm threshold). We consider a light vector mediator with general couplings to SM fermions. For the photon spectra we see a smooth interpolation from typical hadron decay chains with their round spectra down to the pion continuum with a triangular shape. For positrons the main feature is the secondary neutron decay above threshold.

Because we are relying on an updated fit to electron-position input data to HERWIG we can also propagate the uncertainties from poorly measured channels into the photon and positron spectra. Already for relatively heavy dark matter the positron spectrum shows sizeable error bars. In the case of photons, smaller dark matter masses with fewer and less well measured annihilation channels are also plagued by significant error bars, eventually covering an order of magnitude for $m_\chi = 250$ MeV.

Our new implementation closes the gap between standard PYTHIA-based tools such as PPPC4DMID, MICROMEGAS, MADMD, or DARKSUSY and the comparably simple small-mass continuum regime and should allow for a reliable study of GeV-scale dark matter even if it dominantly interacts with SM quarks.*

## Acknowledgments

First, we want to thank Patrick Foldenauer for his early contributions to Figs. 1 and 2. TP is supported by the German Research Foundation DFG under grant no. 396021762-TRR 257 and

---

*The code we have used to produce these results will be available in a future version of HERWIG7. If there is sufficient interest we will also think about providing the output as a cool and fast neural network.

would like to thank Stefan Gieseke for help starting this project. Peter Reimitz is funded by the Graduiertenkolleg *Particle physics beyond the Standard Model* (GRK 1940). Peter Richardson is supported by funding from the UK Science and Technology Facilities Council (grant numbers ST/P000800/1, ST/P001246/1), and benefited from the European Union's Horizon 2020 research and innovation programme as part of the Marie Skłodowska-Curie Innovative Training Network MCnetITN3 (grant agreement no. 722104).

## A    Updated fits with error envelopes

Table 2: Dominant processes contributing to $e^+e^- \to$ hadrons in the relevant energy range.

| Channel | Data | Parametrization | fit | threshold [GeV] |
|---------|------|-----------------|-----|-----------------|
| $\pi\gamma$ | [65] | [65] | [65] | |
| $\pi\pi$ | [66–68] | [69] | [69] | 0.280 |
| $\pi\pi\pi$ | [70] | [71] | [71] | 0.420 |
| $4\pi$ | [72, 73] | [74] | own | 0.560 |
| $\omega\pi$ | [75] | [75] | [75] | 0.918 |
| $p\bar{p}/n\bar{n}$ | [76–93] | [94] | own | 1.877 |
| $\eta\gamma$ | [95] | [95] | [95] | 0.548 |
| $\eta\pi\pi$ | [96, 97] | [98] | own | 0.827 |
| $\eta'\pi\pi$ | [99] | [98] | own | 1.237 |
| $\omega\pi\pi$ | [99–101] | own | own | 1.062 |
| $\eta\phi$ | [102, 103] | own | own | 1.568 |
| $\eta\omega$ | [104] | own | own | 1.331 |
| $\phi\pi$ | [102, 105] | own | own | 1.160 |
| $KK$ | [78, 106–114] | [69] | own | 0.996 |
| $KK\pi$ | [102, 105, 115–117] | own | own | 1.135 |

If we limit ourselves to dark matter annihilation through a vector mediator we can relate the dark matter annihilation process to the corresponding and measurable process

$$e^+e^- \to \text{hadrons.} \tag{10}$$

Its matrix element has the form

$$\mathcal{M} = \frac{e}{\hat{s}}\, \bar{v}_{e^+}\gamma_\mu u_{e^-}\, \langle \text{had}|J^\mu_{\text{em}}|0\rangle \,. \tag{11}$$

The electromagnetic quark current $J^\mu_{\text{em}} = \sum_{q=u,d,s} e_q \bar{q}\gamma^\mu q$ can be decomposed into its isospin components $I = 0, 1$ and its strange-quark content,

$$J^\mu_{\text{em}} = \frac{1}{\sqrt{2}}J^\mu_{I=1,3} + \frac{1}{3\sqrt{2}}J^\mu_{I=0} - \frac{1}{3}J^\mu_s \,, \tag{12}$$

with

$$J^\mu_{I=1,3} = \frac{\bar{u}\gamma_\mu u - \bar{d}\gamma_\mu d}{\sqrt{2}},$$

$$J^\mu_{I=0} = \frac{\bar{u}\gamma_\mu u + \bar{d}\gamma_\mu d}{\sqrt{2}},$$

$$J^\mu_s = \bar{s}\gamma_\mu s \,. \tag{13}$$

Table 3: Parameters of the nucleon form factor from our fit using the model describing $p\,p$ production from Ref. [94].

| | | | | | | | |
|---|---|---|---|---|---|---|---|
| $c_1^{1R}$ | -0.467(12) | $c_1^{1I}$ | -0.385(15) | $c_2^{1R}$ | -0.177(11) | $c_2^{1I}$ | 0.149(12) |
| $c_3^{1R}$ | 0.301(18) | $c_3^{1I}$ | 0.264(16) | $c_1^{2R}$ | 0.052(13) | $c_1^{2I}$ | -3.040(21) |
| $c_2^{2R}$ | -0.003(11) | $c_2^{2I}$ | 2.380(15) | $c_3^{2R}$ | -0.348(11) | $c_3^{2I}$ | -0.104(12) |
| $c_1^{3R}$ | -7.88(47) | $c_1^{3I}$ | 5.67(29) | $c_2^{3R}$ | 10.20(10) | $c_2^{3I}$ | -1.94(31) |
| $c_1^{4R}$ | -0.8320(11) | $c_1^{4I}$ | 0.3080(12) | $c_2^{4R}$ | 0.4050(11) | $c_2^{4I}$ | -0.2500(12) |

We study all hadronic states which appear in the total cross section $\sigma(e^+e^- \to \text{hadrons})$ in the MeV to GeV range. A list of all channels, their parametrizations, their data fits, and their threshold values is given in Tab. 2. Our modelling of the $e^+e^-$ scattering relies on vector meson dominance [118]. In that case the hadronic current $\langle\text{had}|J_{\text{em}}^{\mu}|0\rangle$ can be described by a momentum-dependence and a form-factor that includes all resonances allowed under certain isospin symmetry assumptions. The parametrization and fit values for the form-factors for the $\pi\gamma$, $\pi\pi$, $3\pi$, $\omega\pi$, and $\eta\gamma$ final states are taken from Refs [69, 71, 74], as implemented in the event generator PHOKHARA [119, 120], and the Born cross section formulae from the SND measurements [65, 75, 95]. For all other channels, we provide new fits. Our modelling does not take into account possible final state interactions such as rescattering [121] and Sommerfeld-effects of non-relativistic final states [122]. For example, the K-matrix approach [123] includes such interactions, e.g. the $\pi\pi \leftrightarrow KK$ rescattering above the $KK$ threshold, with an infinite series of rescattering loops. It is used to describe, for example, three-body $B$-decays [124]. The only exception of using rescattering effects is the Flatté parametrization in the $\omega\pi\pi$ channel that takes into account $KK$ threshold effects as seen below.

### $p\bar{p}$ (update)

The data and the fit function for this channel are given in Tab. 2. We updated the data set used for our fit since from the input to the previous fit [94] Ref. [125] is superseded by Ref. [86], Ref. [126] by Ref. [91], and Ref. [127] by Ref. [77]. For asymmetric data uncertainties we symmetrize statistical and systematic uncertainties separately and then add both in quadrature. We refrain from a more sophisticated error analysis for instance including correlations between systematic uncertainties, since in most cases detailed information about the systematic uncertainties is either missing or the statistical uncertainty dominates. For the fit, we get $\chi^2/\text{n.d.f} = 1.069$, and the best-fit values are shown in Tab. 3.

### $\eta\pi\pi, \eta'\pi\pi$ (update)

The fit function for the $\eta\pi\pi$ and $\eta'\pi\pi$ hadronic currents are based on [98]. We re-fit the fit function to more recent data sets [96, 97] compared to those used in [98]. The fit values can be found in Tab. 4.

### $KK$ (update)

We parametrize the hadronic current for the $K^0\bar{K}^0$ and $K^+K^-$ channels in the same way as done in Ref. [69]. Unlike Ref. [69], we do not fix all masses and widths of the $\rho, \omega$ and $\phi$ states to their PDG values but let them float in the fit. Furthermore, we use an updated data set for the fit, as mentioned in Tab. 2 and included the $\tau^- \to K_S^0\pi^-\nu_\tau$ data from Ref. [128] to better constrain the $I = 1$ component of the current. The fit values are listed in Tab. 5. The last coupling of each resonance is calculated via Eq.(16) in Ref. [69], and we keep $\eta_\phi = 1.055$,

$\gamma_\omega = 0.5$ and $\gamma_\phi = 0.2$ fixed such as in Ref. [69]. For the simultaneous fit to $K^0 K^0$ and $K^+ K^-$ data we obtain $\chi^2/\text{n.d.f} = 1.621$.

**$4\pi$ (update)**

For the $4\pi$ channel, we use the parametrization of Ref. [74] and fit it to more recent rate measurements for $e^+ e^- \to 2\pi^0 \pi^+ \pi^-$ and $e^+ e^- \to 2\pi^+ 2\pi^-$ from BaBar [72,73]. We obtain a $\chi^2/\text{n.d.f} = 1.28$ and the fit values are listed in Tab. 6.

**$\eta\phi, \eta\omega, \phi\pi$ (new)**

Our first new fit is to the processes $e^+ e^- \to \eta\phi, \eta\omega, \phi\pi$, where the momentum-dependent Born cross sections are

$$\sigma(s) = \frac{4\pi\alpha_{\text{em}}(s)^2}{3\hat{s}^{3/2}} P_f(s) |F|^2, \tag{14}$$

where $\alpha_{\text{em}}(s)$ is the fine structure constant, $P_f(s) = q^3_{\text{cm},X}$ the final-state phase space, $q_{\text{cm},X}$ the final-state particle momentum and $F$ is the respective form factor. The resonant contributions are simply parametrized by

$$F_{\eta\omega,\eta\phi} = \sum_i \frac{a_i e^{i\varphi_i}}{m_i^2 - \hat{s} - i m_i \Gamma_i},$$

$$F_{\phi\pi} = \sum_i \frac{a_i e^{i\varphi_i}}{m_i^2 - \hat{s} - i \sqrt{\hat{s}} \Gamma(\hat{s})}, \tag{15}$$

where we take the $s$-dependent width $\Gamma(s)$ from Ref. [102]. All parameters and fit values for $\eta\phi$, $\eta\omega$, and $\phi\pi$ production are listed in Tab. 7.

**$\omega\pi\pi$ (new)**

Next, for the $\omega\pi\pi$ channel, we use

$$\langle \omega\pi\pi | J^\mu_{\text{em}} | 0 \rangle = e g^{\mu\nu} \frac{g_{\omega''} m^2_{\omega''}}{\hat{s} - m^2_{\omega''} + i m_{\omega''} \Gamma_{\omega''}} g_{\nu\sigma} \varepsilon^\sigma_\omega \sum_{i=1,2} \text{BW}_{f_i}(q^2) \tag{16}$$

for the hadronic current. In our energy range we only need to consider one vector meson mediator $\omega''$, namely the $\omega(1650)$ meson. For the $f_i$ mediator we have

$$\text{BW}_{f_1}(m_{\pi\pi}) = \frac{g_{\omega''\omega\sigma} m^2_\sigma}{m^2_{\pi\pi} - m^2_\sigma + i m_\sigma \Gamma_\sigma}, \tag{17}$$

Table 4: Fit values for the $\eta\pi\pi$ and $\eta'\pi\pi$ channels.

| Parameter | $\eta\pi\pi$ | $\eta'\pi\pi$ | Parameter | $\eta\pi\pi$ | $\eta'\pi\pi$ |
|---|---|---|---|---|---|
| $m_{\rho_1}$ [GeV] | 1.5400(39) | - | $a_1$ | 0.326(10) | 0 (fixed) |
| $m_{\rho_2}$ [GeV] | 1.7600(58) | - | $a_2$ | 0.0115(31) | 0 (fixed) |
| $m_{\rho_3}$ [GeV] | 2.15 (fixed) | 2.110(36) | $a_3$ | 0 (fixed) | 0.0200(81) |
| $\Gamma_{\rho_1}$ [GeV] | 0.356(17) | - | $\varphi_1$ | $\pi$ (fixed) | - |
| $\Gamma_{\rho_2}$ [GeV] | 0.113(22) | - | $\varphi_2$ | $\pi$ (fixed) | - |
| $\Gamma_{\rho_3}$ [GeV] | 0.32 (fixed) | 0.18(11) | $\varphi_3$ | 0 (fixed) | $\pi$ (fixed) |
| | | | $\chi^2/\text{n.d.f}$ | 0.8732 | 0.9265 |

Table 5: Parameters for the description of $KK$ production from our fit using the model of Ref. [69]. All masses and widths are given in GeV, all other parameters are dimensionless

| | | | | | | | |
|---|---|---|---|---|---|---|---|
| $m_{\rho_0}$ | 0.77549 (PDG) | $\Gamma_{\rho_0}$ | 0.1494 (PDG) | $c_{\rho_0}$ | 1.1149(24) | $c_{\rho_4}$ | -0.0383(66) |
| $m_{\rho_1}$ | 1.5207(53) | $\Gamma_{\rho_1}$ | 0.213(14) | $c_{\rho_1}$ | -0.0504(44) | $c_{\rho_5}$ | 0.0775 (calc.) |
| $m_{\rho_2}$ | 1.7410(38) | $\Gamma_{\rho_2}$ | 0.084(12) | $c_{\rho_2}$ | -0.0149(32) | $\beta_\rho$ | 2.1968 |
| $m_{\rho_3}$ | 1.992(15) | $\Gamma_{\rho_3}$ | 0.290(41) | $c_{\rho_3}$ | -0.0390(45) | - | - |
| $m_{\omega_0}$ | 0.78265 (PDG) | $\Gamma_{\omega_0}$ | 0.00849 (PDG) | $c_{\omega_0}$ | 1.365(44) | $c_{\omega_3}$ | 1.40(27) |
| $m_{\omega_1}$ | 1.4144(71) | $\Gamma_{\omega_1}$ | 0.0854(71) | $c_{\omega_1}$ | -0.0278(83) | $c_{\omega_4}$ | 2.8046 (calc.) |
| $m_{\omega_2}$ | 1.6553(26) | $\Gamma_{\omega_2}$ | 0.1603(26) | $c_{\omega_2}$ | -0.325(30) | $\beta_\omega$ | 2.6936 |
| $m_{\phi_0}$ | 1.0194209(94) | $\Gamma_{\phi_0}$ | 0.004253(21) | $c_{\phi_0}$ | 0.9658(27) | $c_{\phi_3}$ | 0.1653(50) |
| $m_{\phi_1}$ | 1.5948(51) | $\Gamma_{\phi_1}$ | 0.029(18) | $c_{\phi_1}$ | -0.0024(20) | $c_{\phi_4}$ | 0.1195 (calc.) |
| $m_{\phi_2}$ | 2.157(57) | $\Gamma_{\phi_2}$ | 0.67(16) | $c_{\phi_2}$ | -0.1956(19) | $\beta_\phi$ | 1.9452 |

where $m_\sigma$ and $\Gamma_\sigma$ are the mass and width of the $\sigma$ meson and using the Flatté parametrization [130]

$$\mathrm{BW}_{f_0}(m_{\pi\pi}) = \frac{g_{\omega''\omega f_0(980)} m_{f_0(980)} \sqrt{\Gamma_0 \Gamma_{\pi\pi}}}{m_{\pi\pi}^2 - m_{f_0(980)}^2 + i m_{f_0(980)}(\Gamma_{\pi\pi} + \Gamma_{\bar{K}K}^*)}, \tag{18}$$

with

$$\begin{aligned}
\Gamma_{\pi\pi} &= g_{\pi\pi} q_\pi(m_{\pi\pi}), \\
\Gamma_{\bar{K}K} &= \begin{cases} g_{\bar{K}K}\sqrt{(1/4)m_{\pi\pi}^2 - m_K^2}, & \text{above threshold} \\ i g_{\bar{K}K}\sqrt{m_K^2 - (1/4)m_{\pi\pi}^2}, & \text{below threshold} \end{cases} \\
\Gamma_{\bar{K}K}^* &= 0.5 \cdot (\Gamma_{\bar{K}^0 K^0} + \Gamma_{K^+ K^-}), \\
\Gamma_0 &= g_{\pi\pi} q_\pi(m_f)
\end{aligned} \tag{19}$$

for the $f_0(980)$ meson, with parameters from Ref. [131]. If not mentioned otherwise, the parameters are set to their PDG values [129]. The $\sigma$ meson contribution can be viewed as a phase space contribution to the $\omega\pi\pi$ channel more than resonant contribution. Therefore, the width is chosen to be large, see Tab. 8.

Table 6: Parameters for the $4\pi$ channel for our fit using the model from [74]. All masses and widths are in GeV; couplings $\beta_i^j$, ($j = a_1, f_0, \omega$ and $i = 1, 2, 3$) as well as $c_\rho$ are dimensionless; $c_{a_1}$ and $c_{f_0}$ in GeV$^{-2}$ and $c_\omega$ in GeV$^{-1}$.

| | | | | | |
|---|---|---|---|---|---|
| $\bar{m}_{\rho_1}$ | 1.44 (fixed) | $\bar{m}_{\rho_2}$ | 1.74 (fixed) | $\bar{m}_{\rho_3}$ | 2.12 (fixed) |
| $\bar{\Gamma}_{\rho_1}$ | 0.678(18) | $\bar{\Gamma}_{\rho_2}$ | 0.805(29) | $\bar{\Gamma}_{\rho_3}$ | 0.209(29) |
| $\beta_1^{a_1}$ | -0.0519(56) | $\beta_2^{a_1}$ | -0.0416(20) | $\beta_3^{a_1}$ | -0.00189(47) |
| $\beta_1^{f_0}$ | $7.39(0.29)\cdot 10^4$ | $\beta_2^{f_0}$ | $-2.62(0.19)\cdot 10^3$ | $\beta_3^{f_0}$ | 334(87) |
| $\beta_1^\omega$ | -0.367(27) | $\beta_2^\omega$ | 0.036(11) | $\beta_3^\omega$ | -0.00472(77) |
| $c_{a_1}$ | -202.0(24) | $c_{f_0}$ | 124.0(52) | $c_\omega$ | -1.580(73) |
| $c_\rho$ | -2.31(24) | $\chi^2$ | 291 | n.d.f | 228 |

Table 7: Fit values for the $\eta\phi$, $\eta\omega$, and $\phi\pi$ channels.

| Process | $\eta\phi$ | | $\eta\omega$ | | $\phi\pi$ | |
|---|---|---|---|---|---|---|
| $i$ | $\phi'$ | $\phi''$ | $\omega'$ | $\omega''$ | $\rho$ | $\rho'$ |
| $m_i$ [GeV] | $1.67 \pm 0.0063$ | $2.14 \pm 0.012$ | $1.425$ [129] | $1.67 \pm 0.0087$ | $0.77526$ [129] | $1.593$ [102] |
| $\Gamma_i$ [GeV] | $0.122 \pm 0.0075$ | $0.044 \pm 0.033$ | $0.215$ [129] | $0.113 \pm 0.016$ | $0.1491$ [129] | $0.203$ [102] |
| $a_i$ | $0.175 \pm 0.0084$ | $0.0041 \pm 0.0019$ | $0.0862 \pm 0.011$ | $0.0648 \pm 0.0078$ | $0.194 \pm 0.073$ | $0.0214 \pm 0.0035$ |
| $\varphi_i$ | $0$ (fixed) | $2.19 \pm 0.046$ | $0$ [104] | $\pi$ [104] | $0$ (fixed) | $121 \pm 16.9$ deg. |
| $\chi^2$/n.d.f | $0.9388$ | | $1.3332$ | | $0.9798$ | |

### $KK\pi$ (new)

Below 2 GeV center-of-mass energy the process $e^+e^- \to KK\pi$ is dominated by $e^+e^- \to KK^* \to K(K\pi)$ where $KK^*$ can be either $K^0 K^{*0}(890)$ or $K^\pm K^{*\mp}(890)$. We can relate the possible final states through their isospin $I = 0, 1$ and can use the following relations for the corresponding amplitudes $A_{0,1}$ [132],

$$
\begin{aligned}
K^+(K^-\pi^0) + K^-(K^+\pi^0) : & \quad \frac{1}{\sqrt{6}}(A_0 - A_1), \\
K_S^0(K_L^0\pi^0) + K_L^0(K_S^0\pi^0) : & \quad \frac{1}{\sqrt{6}}(A_0 + A_1), \\
K^0(K^-\pi^+) + \bar{K}^0(K^+\pi^-) : & \quad \frac{1}{\sqrt{3}}(A_0 + A_1), \\
K^+(\bar{K}^0\pi^-) + K^-(K^0\pi^+) : & \quad \frac{1}{\sqrt{3}}(A_0 - A_1) .
\end{aligned}
\tag{20}
$$

For the amplitudes with intermediate resonances, $e^+e^- \to V \to KK^*$, we use the standard Breit-Wigner dsitribution

$$
A_I = \sum_i A_{I,i} \frac{m_{I,i}^2 e^{\varphi_{I,i}}}{m_{I,i}^2 - \hat{s} - i\sqrt{\hat{s}}\Gamma_{I,i}} .
\tag{21}
$$

In the energy range we are dealing with, we expect the resonances to be $\phi(1680)$ and $\phi(2170)$ for $I = 0$ and $\rho(1450)$ and $\rho(1700)$ for $I = 1$. The lower resonances $\rho(770)$ and $\phi(1020)$ are not considered in the energy range of the fit and we set their couplings to zero. Furthermore,

Table 8: Fit values for the $\omega\pi\pi$ channel.

| Parameter | Fit value | PDG |
|---|---|---|
| $m_{\omega''}$ | $1.69 \pm 0.00919$ GeV | $1.670 \pm 0.03$ GeV |
| $\Gamma_{\omega''}$ | $0.285 \pm 0.0143$ GeV | $0.315 \pm 0.035$ GeV |
| $m_\sigma$ | $0.6$ GeV | - |
| $\Gamma_\sigma$ | $1.0$ GeV | - |
| $g_{\omega''\omega\sigma}$ | $1.$ (fixed) | - |
| $m_{f_0(980)}$ | $0.980$ GeV | $0.990 \pm 0.020$ GeV |
| $\Gamma_{f_0(980)}$ | $0.1$ GeV | $0.01$-$0.1$ GeV |
| $g_{\omega''\omega f_0(980)}$ | $0.883 \pm 0.0616$ | - |
| $g_{\omega''}$ | $1.63 \pm 0.0598$ | - |
| $\chi^2$/n.d.f | $2.001$ | |

Table 9: Fit values for the $KK\pi$ channel.

| fit value | $I$ | $i = 1$ | $i = 2$ | $i = 3$ |
|---|---|---|---|---|
| $A_{I,i}$ in GeV$^{-1}$ | $I = 0$ | 0 (fixed) | $0.233 \pm 0.020$ | $0.0405 \pm 0.0081$ |
| | $I = 1$ | $-2.34 \pm 0.15$ | $0.594 \pm 0.023$ | $-0.018 \pm 0.013$ |
| $\varphi_{I,i}$ | $I = 0$ | 0 (fixed) | $1.1\text{E-}07 \pm 0.092$ | $5.19 \pm 0.34$ |
| | $I = 1$ | 0 (fixed) | $0.317 \pm 0.056$ | $2.57 \pm 0.32$ |
| $m_{I,i}$ [GeV] | $I = 0$ | 1.019461 (fixed) | $1.6334 \pm 0.0065$ | $1.957 \pm 0.034$ |
| | $I = 1$ | 0.77526 (fixed) | 1.465 (fixed) | 1.720 (fixed) |
| $\Gamma_{I,i}$ [GeV] | $I = 0$ | 0.004249 (fixed) | $0.218 \pm 0.013$ | $0.267 \pm 0.032$ |
| | $I = 1$ | 0.1491 (fixed) | 0.400 (fixed) | 0.250 (fixed) |

we fix the mass and the width of the intermediate $K^*$ resonance to $m_{K^*} = 0.8956$ GeV and $\Gamma_{K^*} = 0.047$ GeV and use a $p$-wave Breit-Wigner propagator of the form

$$\text{BW}_{K^*}(s) = \frac{g_{K^*K\pi} m_{K^*}^2}{m_{K^*}^2 - s - i\sqrt{s}\Gamma(s)}, \tag{22}$$

with the $s$-dependent width

$$\Gamma(s) = \Gamma_{K^*} \frac{\sqrt{s}}{m_{K^*}} \left( \frac{\beta(s, m_1, m_2)^2}{\beta(m_{K^*}, m_1, m_2)^2} \right)^{3/2}, \tag{23}$$

where $m_1, m_2$ are the decay products of the $K^*$ state and

$$\beta(s, m_1, m_2) = \left( 1 - \frac{(m_1 + m_2)^2}{s} \right)^{1/2} \left( 1 - \frac{(m_1 - m_2)^2}{s} \right)^{1/2} \tag{24}$$

is their velocity in the rest frame of $K^*$. The $K^* K \pi$ coupling is given by

$$g_{K^*K\pi} = \sqrt{6\pi m_{K^*}^2 / (0.5 m_{K^*} \beta(m_{K^*}^2, m_{K^\pm}, m_{\pi^\pm}))^3 \Gamma_{K^*}} = 5.37392360229 \,. \tag{25}$$

Furthermore, we include a small $\phi\pi^0$ contribution for final states including neutral pions by adding the $\phi\pi^0$ cross section obtained by the $\phi\pi$ fit and the corresponding branching fractions $\text{BR}(\phi(1020) \rightarrow K_L^0 K_S^0) = 0.342$ and $\text{BR}(\phi(1020) \rightarrow K^+ K^-) = 0.489$. We perform a simultaneous fit to all possible final states in order to obtain the fit parameters of the amplitudes $A_{0,1}$. The fit values can be found in Tab. 9.

We show all numerical best-fit solutions as blue lines for all final states in Figs. 6, 7, and 8. The error bars on the data are dominated by statistical uncertainties. All fits describe the most recent data sets over the entire range shown.

**Error bands**

In addition to the central values of the relevant parameters describing the $e^+e^-$ data we also estimate the error bands for the relevant processes. The reason is that some of the channels are rather poorly measured, and it is important to propagate these uncertainties through the analysis. Because most fit parameters are physical parameters appearing in the analytic description of the $e^+e^-$ cross sections, such as masses or widths or rates, we do not find them suitable for a proper statistical analysis. For instance a total cross section measurement will lead to uncontrolled correlations between widely different phase space regions in the fit, where

the different phase space regions are crucial to describe the dark matter spectra for a variable dark matter mass. Examples for the impact of a known form of the energy dependence of the scattering process on poorly measured phase space regions are the $\eta\pi\pi$ channel in Fig. 6, the $\pi\pi$ channel in Fig. 7, or the $3\pi$ channel in Fig. 8.

Instead, we define envelopes by varying a sub-set of fit parameters around their mean value within their uncertainty provided our python IMINUIT [133, 134] fit or as stated in papers. For poorly resolved peak structures as in the $\eta'\pi\pi$, $\phi\pi$, and $\eta\omega$ case or higher resonances as in $\eta\phi$ and $KK\pi$, we do not vary any widths and only some masses, since they are determined from the peak structure and bias the off-peak spectrum through correlations. The contribution of phases to our envelopes is only considered if no other set of parameters is sufficient to describe the measurement uncertainties. For channels with simple parametrizations with fixed masses and widths and floating peak cross sections and phases as in the case of $\pi\gamma$ [65] and $\eta\gamma$ [95], we vary all peak cross sections and the phases of the $\phi$ and $\omega$ resonance, respectively. In these cases, we see that away from the resonance region the error envelopes increase. For precisely measured phase space regions, we consider the full set of parameters describing these regions. These are usually large peak structures such as the $\phi \to KK$ and $\rho \to \pi\pi$ resonances in Fig. 7 or the $\omega, \rho \to 3\pi$ peak around 0.78 GeV in Fig 8. Those resolved regions turn out to be well described and are stable against variations of the parameters, so they give only small envelopes.

It can be challenging or nearly impossible to obtain consistent envelopes for some channels, where one parametrization is used for several sub-channels as in the case of $KK$ and $p\bar{p}/n\bar{n}$. As long as the shape of the data is the same as in the case of $4\pi$, $KK\pi$ and the $\phi$ resonance region in the $KK$ channel, this does not cause any problems. Here we can assume that a parameter and its variation influence the fit curve in the same way. However, for energies above 1.4 GeV in the $KK$ channel, the trend of the data of $K^+K^-$ and $K^0\bar{K}^0$ is completely different. Therefore, already the fit to the data is challenging and only possible by allowing for more resonance fit parameters in the parametrization [69]. A variation of the parameter might influence both channels differently and it is not clear that an extremal value in the one case is also extremal in the other. This tension of both data sets causes too small error bands for energies above 1.8 GeV. For the $p\bar{p}/n\bar{n}$ channel, we do not have sufficient data for $n\bar{n}$ to describe this channel properly as already described in Ref. [94].

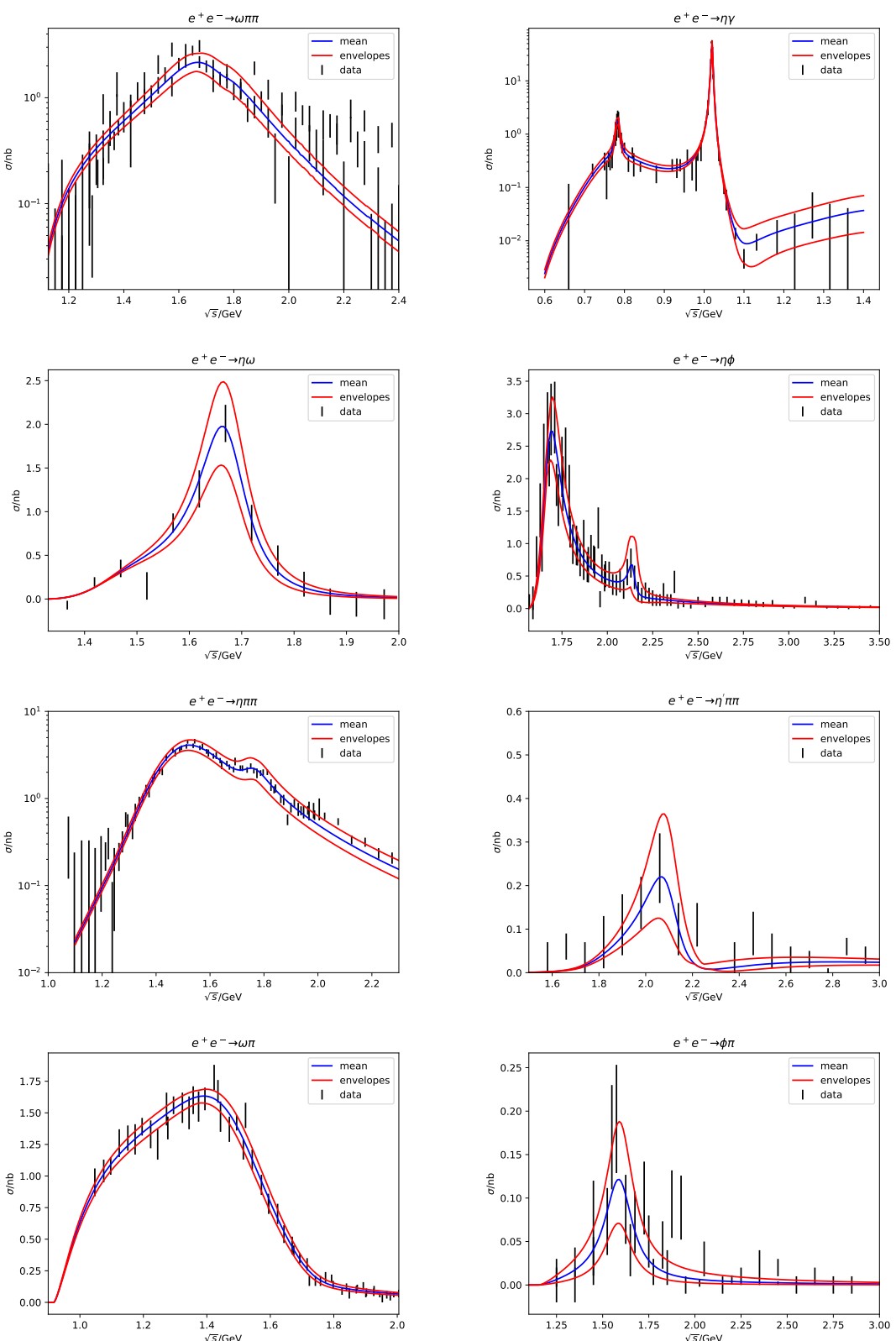

Figure 6: Cross sections for hadronic final states with error envelopes.

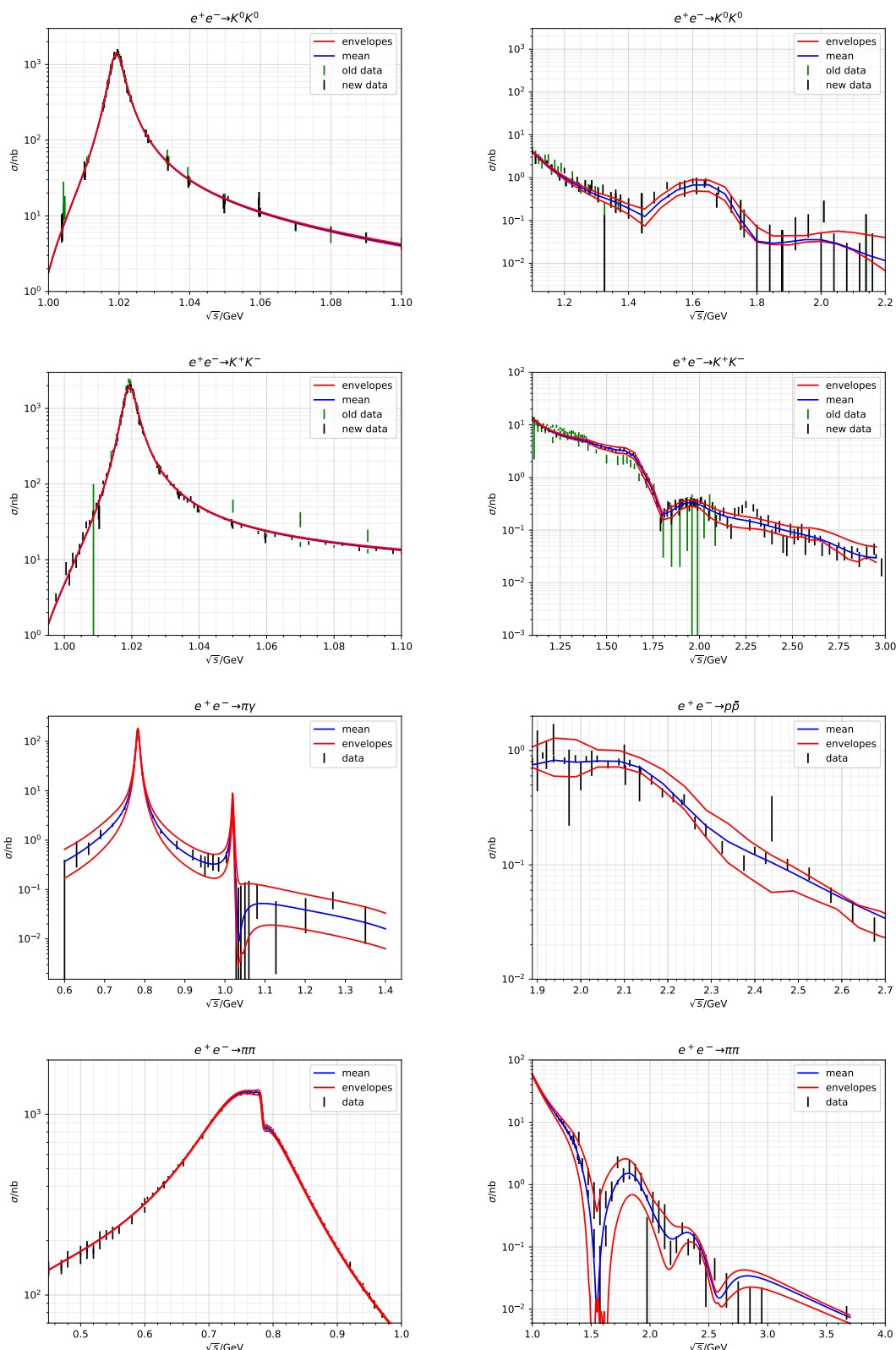

Figure 7: Cross sections for hadronic final states with error envelopes.

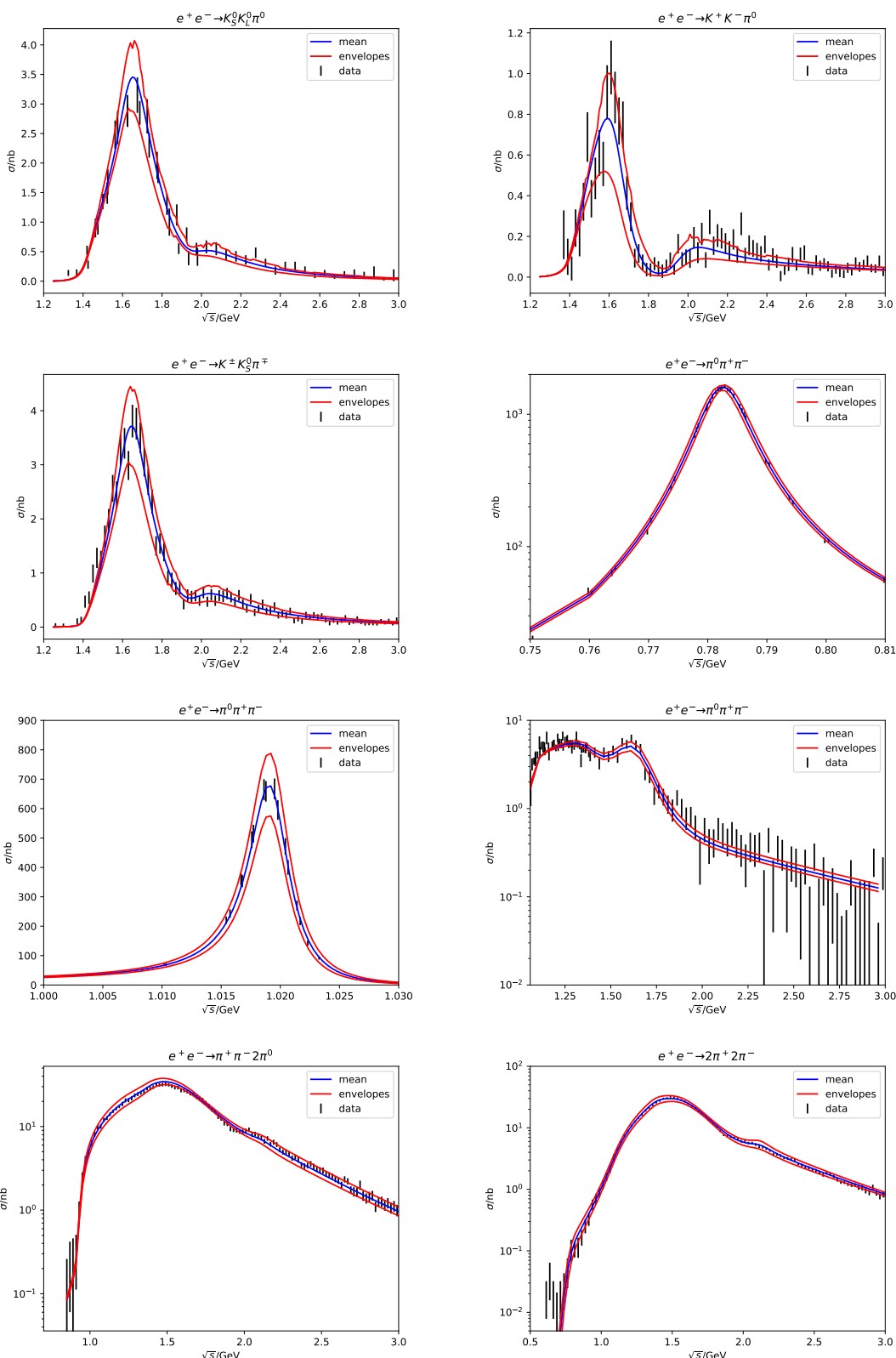

Figure 8: cross sections for hadronic final states with error envelopes.

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
