# Peer review of "Hadronic Footprint of GeV-Mass Dark Matter"

_SciPost Physics, doi:SciPost Phys. 8, 092 (2020)_

## Round 2 · Referee Report · Anonymous (Referee 1) · 2020-4-20

Report

The authors present an update of HERWIG7 that allows to calculate the photon and positron spectra produced in the annihilation of Majorana dark matter particles through a light vector mediator. They focus in the mass range between 250 MeV and 5 GeV, where the results from the most popular dark matter tools start to lose validity. Using the latest results for various hadronic currents, the authors calculate the photon and positron spectra for various annihilation final states, and they find significant differences with respect to the ones obtained with the published tools. The analysis is solid, the paper is well written ad the results are of relevance for indirect dark matter searches. In view of this, I am happy to recommend publication of the manuscript in the present form.

---

## Round 2 · Referee Report · Anonymous (Referee 2) · 2020-4-29

Strengths

Very interesting paper filling the gap in what the DM annihilation spectrum might look like for masses between 250 MeV and 5 GeV

Weaknesses

  1. on page 3-4, in the discussion of "weakening the CMB limits" the discussion should tackle the issue of how one can weaken those limits while at the same time giving some indirect detection signal - the examples quoted, such as asymmetric dark matter, are pointless.

  2. Statements such as "This allows us to cover DM masses down to twice the pion mass" should be better qualified: the authors limit themselves to only one mediator case, vector mediator, and they should clarify that their results and methods are intrinsically limited to that case, unlike other tools that are largely mediator-independent.

  3. There should be a discussion of possible final-state interaction effects

  4. There are a few typos (e.g. on page 14 "An list of all channels")

Report

Very interesting paper filling the gap in what the DM annihilation spectrum might look like for masses between 250 MeV and 5 GeV.

I have a few points that need to be addressed in a revised version:

  1. on page 3-4, in the discussion of "weakening the CMB limits" the discussion should tackle the issue of how one can weaken those limits while at the same time giving some indirect detection signal - the examples quoted, such as asymmetric dark matter, are pointless.

  2. Statements such as "This allows us to cover DM masses down to twice the pion mass" should be better qualified: the authors limit themselves to only one mediator case, vector mediator, and they should clarify that their results and methods are intrinsically limited to that case, unlike other tools that are largely mediator-independent.

  3. There should be a discussion of possible final-state interaction effects

  4. There are a few typos (e.g. on page 14 "An list of all channels")

Requested changes

See above

  • validity: high
  • significance: high
  • originality: top
  • clarity: high
  • formatting: excellent
  • grammar: reasonable

Author:  Peter Reimitz  on 2020-06-08  [id 851]

(in reply to Report 2 on 2020-04-29)
Category:
correction

Changes made in the new version of the paper:

  1. In section, we explain in more detail what we mean by evading CMB bounds with asymmetric dark matter while still having an indirect detection signal. We refer to a paper from Kathryn Zurren for more details.

  2. We made it clearer in the text that we only focus on vector mediators, e.g. in the abstract, in section 2 where we explain why we choose these models, and in the sentence quoted by the referee above. We also mention in the beginning of section 3 that the established tools are mediator-independent.

  3. In the first part of the Appendix, we discuss possible final-state interaction and that they are not included in our approach.

  4. We tried to correct some more typos.

---

## Round 3 · Referee Report · Anonymous (Referee 2) · 2020-6-9

Strengths

The authors have adequately addressed the points I raised in my previous report. I recommend publication.

Weaknesses

N/A

Report

The authors have adequately addressed the points I raised in my previous report. I recommend publication.

---

## Editorial Decision

published